# Impact of six-month COVID-19 travel moratorium on *Plasmodium falciparum* prevalence on Bioko Island, Equatorial Guinea

Dianna E. B. Hergott [1,2] ✉, Carlos A. Guerra [3], Guillermo A. García [3], Jeremías Nzamío Mba Eyono [4], Olivier T. Donfack[4], Marcos Mbulito Iyanga[4], Restituto Mba Nguema Avue[4], Crisantos Nsue Abeso Nsegue[4], Teresa Ayingono Ondo Mifumu[4], Matilde Riloha Rivas[5], Wonder P. Phiri[4], Sean C. Murphy [6,7,8], Brandon L. Guthrie[1,9], David L. Smith [2,10] & Jennifer E. Balkus [1,11]

Importation of malaria infections is a suspected driver of sustained malaria prevalence on areas of Bioko Island, Equatorial Guinea. Quantifying the impact of imported infections is difficult because of the dynamic nature of the disease and complexity of designing a randomized trial. We leverage a six-month travel moratorium in and out of Bioko Island during the initial COVID-19 pandemic response to evaluate the contribution of imported infections to malaria prevalence on Bioko Island. Using a difference in differences design and data from island wide household surveys conducted before (2019) and after (2020) the travel moratorium, we compare the change in prevalence between areas of low historical travel to those with high historical travel. Here, we report that in the absence of a travel moratorium, the prevalence of infection in high travel areas was expected to be 9% higher than observed, highlighting the importance of control measures that target imported infections.

Despite increased efforts and control strategies, *Plasmodium falciparum (Pf)* malaria remains endemic in 85 countries and territories[1]. The increased frequency in which individuals move between and within countries has created added challenges for areas that have recently eliminated malaria, as well as those working towards elimination[2–10]. Bioko Island, Equatorial Guinea, has seen a significant decrease in malaria burden over the past two decades[11]; however, there are several areas of the island, especially urban areas where prevalence is typically lower, that have not yet approached pre-elimination levels. These urban areas also tend to have a higher proportion of individuals

[1]Department of Epidemiology, School of Public Health, University of Washington, Seattle, Washington, USA. [2]Institute for Health Metrics and Evaluation, University of Washington, Seattle, Washington, USA. [3]MCD Global Health, Bioko Island Malaria Elimination Project, Silver Spring, Maryland, USA. [4]MCD Global Health, Bioko Island Malaria Elimination Project, Malabo, Equatorial Guinea. [5]National Malaria Control Program, Ministry of Health and Social Welfare, Malabo, Equatorial Guinea. [6]Department of Laboratory Medicine and Pathology, University of Washington, Seattle, Washington, USA. [7]Department of Microbiology, University of Washington, Seattle, Washington, USA. [8]Department of Laboratories, Seattle Children's Hospital, Seattle, Washington, USA. [9]Department of Global Health, School of Public Health, University of Washington, Seattle, Washington, USA. [10]Department of Health Metrics Science, University of Washington, Seattle, Washington, USA. [11]Present address: Public Health-Seattle & King County, Seattle, Washington, USA. ✉e-mail: dhergott@uw.edu

who travel between Bioko Island and the mainland of Equatorial Guinea[12], where malaria prevalence is substantially higher[7]. Several previous studies suggest that there is a high amount of importation of malaria to Bioko Island in returning travelers[13], and that these imported infections contribute to sustained prevalence in urban areas[7,14]. However, previous analyses are based on retrospective reporting of travel history captured via cross-sectional surveys, and there have been no studies that have allowed for direct estimation of the impact of imported malaria cases to prevalence in high travel areas. Better estimates of the contribution of imported cases to malaria transmission are needed to inform malaria control measures.

In 2020, to address the COVID-19 pandemic and minimize local transmission of SARS-CoV-2, Equatorial Guinea imposed travel restrictions throughout the country, eliminating movement between the islands and the mainland from March to September 2020[15]. This, ostensibly, also eliminated the importation of *Plasmodium* infections to Bioko. The travel restriction provides a natural experiment in which the impact of imported infections can be directly assessed. Here, we compare the prevalence and odds of malaria infection before and after the travel restriction in areas that historically have had a high volume of travel to areas of historically low volume of travel using a difference in differences analysis. This allows us to directly assess the impact of imported malaria infection on prevalence on Bioko Island.

Data on malaria infection was collected before (2019) and after (2020) the travel moratorium through a household-based malaria indicator survey (MIS) with households selected from the whole island[16,17]. Using historical reported travel from MIS between 2015 and 2018, we use the distribution of travel frequency by enumeration area (EA) to classify areas as high travel (in the top quartile) or low travel (bottom quartile). In this work, we then compare the change in malaria prevalence and odds of infection in high and low travel areas before and after the travel mortarium to estimate the contribution of imported malaria to prevalence in high travel areas.

## Results

### Travel prevalence and sample selection
In 2019 and 2020, there were 109 EAs sampled in the MIS. Based on 2015–2018 data, the range of travel prevalence to the mainland of Equatorial Guinea in each EA, defined as the smoothed proportion of individuals who had traveled in the past eight weeks[7], ranged from 1.5% to 39.9%. We classified the EAs in the top quartile of the travel distribution ($\geq 12.2\%$) as high travel areas, and those in the bottom quartile of the travel distribution ($\leq 4.3\%$) as low travel areas, resulting in 56 EAs classified as high or low travel areas included in this analysis (Fig. 1). The distribution of households and individuals in each of the travel classifications is shown in Table 1. Low travel areas had fewer individuals sampled compared to high travel areas, consistent with the population densities of these areas. Analyses were done using data from 12,128 of the 13,195 (92%) individuals who had non-missing data for all variables in the final analysis. There were no noticeable differences in the distribution of covariates in the full data set compared to the analytic data set.

### Malaria prevalence before and after travel restrictions
The unadjusted prevalence of malaria in 2019 was 7.3% in low travel areas (95% CI: 4.5, 10.1), and 13.6% in high travel areas (95% CI: 12.4, 14.9). In 2020, prevalence in low travel areas increased to 12.8% (95% CI: 7.2, 18.5) while decreasing to 11.8% in high travel areas (95% CI: 10.0, 13.5), representing a prevalence difference of 5.5% (95% CI: 0.9, 10.1), and −1.9% (95% CI: −3.2, −0.5), respectively (Table 2). Assuming parallel trends in high and low travel areas, the *Pf* prevalence difference in high travel areas was 7.4% lower than would have been expected in the absence of the travel moratorium (95% CI: −12.1%, −2.6%), without adjustment.

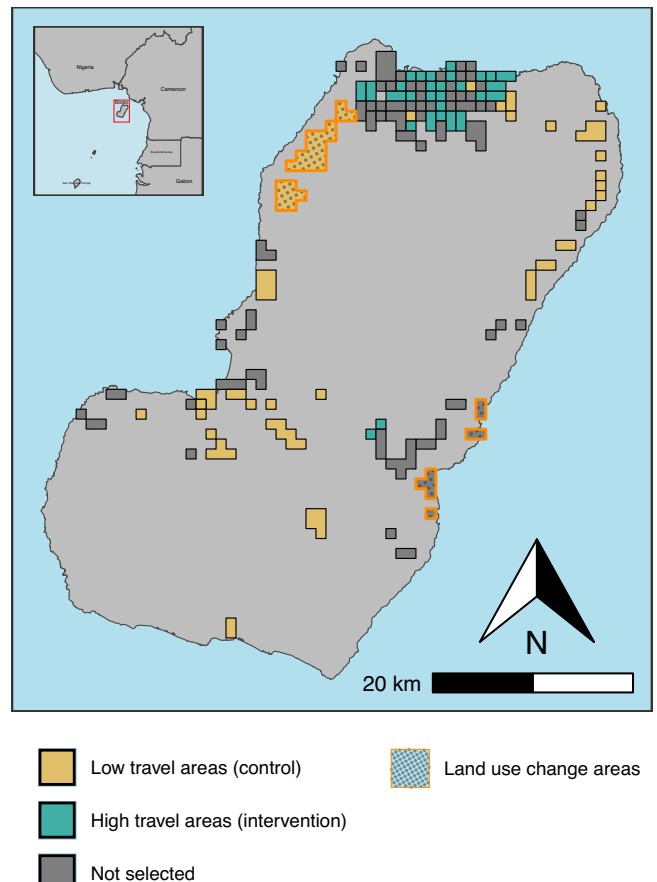

**Fig. 1 | Map of Bioko Island, Equatorial Guinea, showing enumeration areas (EAs) selected for the difference in differences analysis.** EAs in teal are those in the top quartile of historical smoothed travel prevalence. EAs in gold are those in the bottom quartile of historical smoothed travel prevalence. EAs in gray were not selected for the analysis. EAs outlined in orange and filled with patterned dots are the areas with known land use changes over the study period.

A variety of factors known to be related to malaria risk were compared between 2019 and 2020 within each of the travel groups (Supplementary Table 1). Of those evaluated, time that individuals went indoors, insecticide spray coverage, within island travel, and presence of air conditioning showed significant differences between years and were included in the final model. The proportion of individuals who reported going inside their house before 7 pm increased 2.5% in low travel areas (22.5% vs. 25.1%) and 6% in high travel areas (29.2% vs. 35.4%). The proportion of households sprayed with insecticide decreased in 2020 (35.1%) compared to 2019 (46.9%) in low travel areas, whereas the spray coverage in high travel areas was higher in 2020 compared to 2019 (57.2% vs. 28.3%), consistent with the targeted IRS approach of the program[18]. The proportion of respondents who indicated they traveled within the island (and spent at least one night away from their home) in the past eight weeks decreased from 2019 to 2020 in both low travel areas (18.8% vs. 13.3%) and high travel areas (13.2% vs. 9.1%). The proportion of households with air-conditioning increased 6% in high travel areas from 2019 to 2020 (32.7% vs. 38.5%). There were also significant decreases in care-seeking behavior among respondents who were sick from 2019 to 2020 in both low travel areas (52.1% vs. 45.2%) and high travel areas (68.4% vs. 58.4%). However, given that this variable was only available for a small fraction of respondents, inclusion of this covariate was only included in sensitivity analyses only.

When adjusting for spray coverage, going inside before 7 pm, within island travel, and air-conditioning, prevalence in low travel areas

**Table 1 | Total enumeration areas (EA), households, and individuals sampled in the Bioko Island Malaria Indicator Survey and included in the difference in differences analysis, stratified by travel area and stratum for 2019 and 2020**

| | Full Data Set | | | | | | Analytic Data Set | | | | | |
|---|---|---|---|---|---|---|---|---|---|---|---|---|
| | Low Travel Areas | | | High Travel Areas | | | Low Travel Areas | | | High Travel Areas | | |
| | Stratum 1 n(%) | Stratum 2 n(%) | Total n | Stratum 1 n(%) | Stratum 2 n(%) | Total n | Stratum 1 n(%) | Stratum 2 n(%) | Total n | Stratum 1 n(%) | Stratum 2 n(%) | Total n |
| Enumeration Areas | 17 (0.61) | 11 (0.39) | 28 | 5 (0.18) | 23 (0.82) | 28 | 17 (0.61) | 11 (0.39) | 28 | 5 (0.18) | 23 (0.82) | 28 |
| 2019 | | | | | | | | | | | | |
| Households Sampled | 498 (0.62) | 305 (0.38) | 803 | 179 (0.12) | 1356 (0.88) | 1535 | 489 (0.62) | 301 (0.38) | 790 | 177 (0.12) | 1337 (0.88) | 1514 |
| Individuals Sampled | 1334 (0.57) | 1021 (0.43) | 2355 | 457 (0.1) | 4079 (0.9) | 4536 | 1288 (0.57) | 980 (0.43) | 2268 | 431 (0.1) | 3818 (0.9) | 4249 |
| 2020 | | | | | | | | | | | | |
| Households Sampled | 468 (0.64) | 268 (0.36) | 736 | 147 (0.11) | 1226 (0.89) | 1373 | 438 (0.63) | 255 (0.37) | 693 | 136 (0.1) | 1178 (0.9) | 1314 |
| Individuals Sampled | 1110 (0.57) | 848 (0.43) | 1958 | 450 (0.1) | 3896 (0.9) | 4346 | 987 (0.57) | 734 (0.43) | 1721 | 397 (0.1) | 3493 (0.9) | 3890 |

The full data set includes all individuals and their respective households who were tested for *Plasmodium falciparum* during the survey. The analytic data set includes all individuals and their respective households who had non-missing data for all variables included in the adjusted analysis.

**Table 2 | Number of individuals tested (n), *Plasmodium falciparum (Pf)* prevalence estimate and 95% confidence interval (CI) by year and travel group**

| Group and Year | Individuals Tested (n) | Pf Prevalence (95% CI) | Difference in Prevalence (95% CI) | | Difference in Differences (95% CI) | |
|---|---|---|---|---|---|---|
| | | | Unadjusted | Adjusted‡ | Unadjusted | Adjusted‡ |
| Low travel -2019 | 2268 | 7.3% (4.5, 10.1) | 5.5 (0.9,10.1) | 5.8 (0.5,11) | −7.4 (−12.1,−2.6) | −9.2 (−14.5,−3.9) |
| Low travel -2020 | 1721 | 12.8% (7.2, 18.5) | | | | |
| High travel -2019 | 4249 | 13.6% (12.4, 14.9) | −1.9 (−3.2,−0.5) | −3.4 (−5.1,−1.8) | | |
| High travel -2020 | 3890 | 11.8% (10.0, 13.5) | | | | |

Difference in prevalence compares 2020 to 2019 prevalence by travel area. The difference in differences compares to two prevalence differences. Differences are presented for the adjusted and unadjusted models. Estimates and CIs are derived from linear combinations of the outputs from a design-based linear regression model.
‡Adjusted for coverage of Indoor Residual Spraying (IRS), going indoors before 7PM, within island travel, and air conditioning.

increased by 5.8% (95% CI:0.5, 11.0) between 2019 and 2020, but decreased in high travel areas by 3.4% (95% CI: −5.1, −1.8) over the same period. After adjustment, the *Pf* prevalence in high travel areas was 9.2% lower than would have been expected in the absence of the travel moratorium (95% CI: −14.7%, −3.7%). Full model outputs are available in Supplementary Table 2.

Similar results were seen when evaluating the relationship on a relative scale. Comparing the change from 2019 to 2020 in high travel areas to low travel areas, the unadjusted odds of *Pf* infection after the travel moratorium were 55% lower in high travel areas (OR = 0.45, 95% CI: 0.29, 0.71) than would be expected. Following adjustment, odds of infection was 62% lower in high travel areas compared to what would have been expected based on trends in low travel areas (aOR=0.38; 95% CI: 0.22, 0.63). Full results are shown in Supplementary Table 3 and Supplementary Fig. 1.

**Sensitivity analyses**

There were three areas on the island, composed of seven EAs, with known land use changes over the study period that may have impacted risk of malaria transmission, outlined by orange in Fig. 1. Three of these EAs on the western coast were included in our analysis. Evaluation of estimated prevalence in 2019 and 2020 in these three areas showed that in two of the three EAs, the observed prevalence difference between years was greater than the average difference in the low travel group (Supplementary Table 4). When we removed these three EAs from the analysis, the *Pf* prevalence difference in high travel areas was 5.4% lower than would have been expected in the absence of the travel moratorium (95% CI: −9.0, −1.7) in the unadjusted analysis and 6.8%

lower than expected in the adjusted analysis (95% CI: −10.5, −3.1) (Supplementary Table 5). Comparing the relative change from 2019 to 2020 in high travel areas to low travel areas, the adjusted odds of *Pf* infection after the travel moratorium were 54% lower in high travel areas (aOR= 0.44, 95% CI: 0.26, 0.76) (Supplementary Table 3).

There was a decrease in care-seeking behavior in both travel groups from 2019 to 2020, which may result in fewer malaria infections being cleared. To evaluate the possible impact of this behavior, we created an EA level variable for care seeking, calculated as the proportion of those who were sick that sought care within each EA. Adding this variable to the model did not change the outcome: after adjustment, the *Pf* prevalence in high travel areas was 9.3% lower than would have been expected in the absence of the travel moratorium (95% CI: −14.5%, −4.0%). (Supplementary Table 6).

Given that most high travel areas are more urban (stratum 2) and low travel areas are more rural (stratum 1), we ran the analysis separately by stratum and conducted an analysis on a subset of high travel rural areas (n = 5) compared to low travel urban areas (n = 11) to verify that the observed changes were due to travel and not other differences between urban and rural areas over the analysis period. In stratified analyses, the *Pf* prevalence difference in was slightly greater rural areas (−10.4%, 95% CI: −19.5%, −1.2%) compared to urban areas (−8.3%, 95% CI: −14.8%, −1.8%), but the overall trend remained (Supplementary Table 7). In the subset analysis, results were similar to the full data set; prevalence in high travel areas decreased 6.9% (95% CI: −15%, 1.5%) more than what was expected in the unadjusted model and 8.5% (95% CI: −18.1%, 1.9%) more than expected in the fully adjusted model.

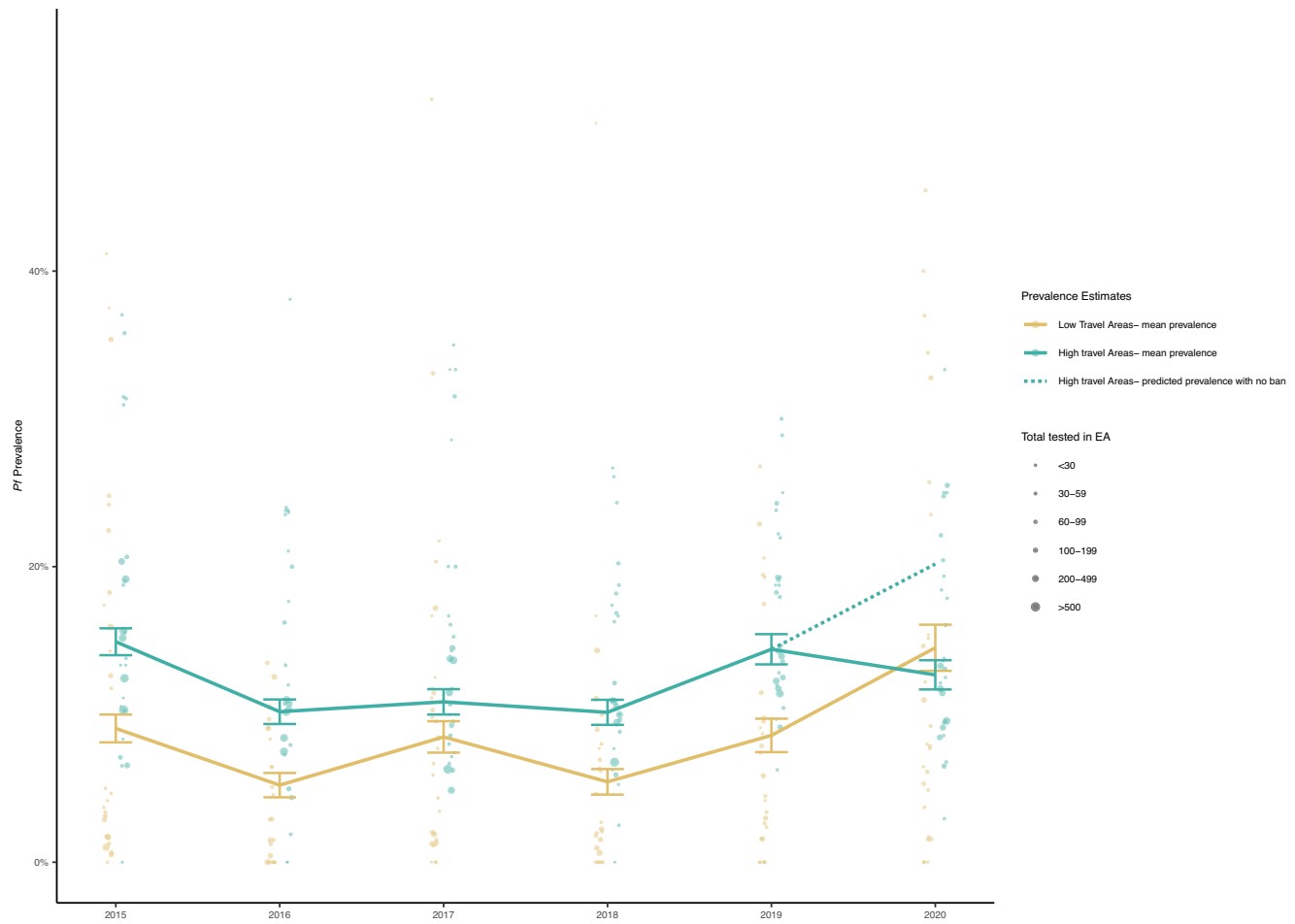

**Fig. 2 | Test of parallel trends assumption, showing the estimated mean *Plasmodium falciparum (Pf)* prevalence in low travel (gold) and high travel (teal) areas from 2015 to 2020.** The 95% confidence intervals (CI) from the grouped mean estimates are shown at each point estimate. The dashed line represents predicted prevalence in 2020 in a counterfactual scenario with no travel moratorium. Prevalence estimates from each enumeration area (EA) in each year are presented as points, and their size is proportional to the number of individuals tested within the EA. The total number of individuals included in the calculation for each year is as follows: 2015: high ($n = 5889$), low ($n = 3560$); 2016: high ($n = 5097$), low($n = 2799$); 2017: high ($n = 5050$), low($n = 2642$); 2018: high ($n = 4861$), low ($n = 2647$); 2019: high ($n = 4532$), low($n = 2354$); 2020: high ($n = 4346$), low ($n = 1957$).

## Model Validity

For the results of a difference in differences model to be valid, the parallel trends assumption must hold[19]. Using data from 2015-2018, the mean prevalence in high and low travel EAs was calculated and plotted with unweighted data from 2019 and 2020. Figure 2 indicates that the trend from 2015 to 2019 was similar in low and high travel groups, and then diverged in 2020 when travel was stopped. A generalized linear mixed effects model with individual level data of the *Pf* positivity for 2015 to 2019 among high and low travel areas, with an interaction term for each year and travel group showed a consistent difference in prevalence of around 7% between high and low travel areas. There was no significant difference in prevalence difference between the two areas except for 2017, when the difference in prevalence between the two areas was 2.7% lower than the difference in 2015 ($p = 0.005$) (Supplementary Table 8). This divergence from trend may have been due to increased construction in several areas in the low travel group during this year. Removing those two areas from the model resulted in a decreased difference (1.9%) which was no longer significant ($p = 0.052$). The model was repeated using a random sample of 80% of the data for 50 iterations. The interaction in 2017 was significant in almost all the iterations (39/50). All other years showed significant difference in <8% of iterations.

## Discussion

Simulation models suggest there are areas of Bioko Island in which high proportions of malaria prevalence can be attributed to infections acquired while traveling to higher burden areas[7,13,14]. A recent model suggested that drastically reducing the number of imported infections between Bioko Island and the mainland could significantly reduce prevalence in areas with a high proportion of travelers[14]. However, prior to 2020, there were, understandably, no intervention studies nor other data to definitively support the model simulations. Travel restrictions imposed as a measure to control the spread of SARS-CoV-2 in 2020 provided an opportunity to directly evaluate the impact of imported infections. We observed that in the absence of travel, prevalence in historically high travel areas decreased by 2%, while prevalence in low travel areas increased by 5% over the same period. This suggests that, assuming parallel trends, in the absence of the travel moratorium, one would have expected *Pf* prevalence to be 7% higher in high travel areas than what was observed. When adjusting for spray coverage, time one went indoors, within island travel, and air conditioning, the difference in trends increased to 9% and remained significant.

In 2019, prior to the travel restrictions, odds of malaria infection were two to three times higher in areas of Bioko Island with a historically high proportion of travelers, relative to areas of historically low

travel. This finding is similar to a 2013 analysis that showed infection risk was greater in children living in areas with the highest proportion of travelers[13]. In 2020, when the movement of individuals was restricted, there was no difference in risk of malaria infection observed in high travel areas compared to low travel areas. This observation both further supports the hypothesis that a significant fraction of the *Pf* prevalence observed in high travel areas could be explained by imported infections[7], while also suggesting that malaria risk in these areas is not solely driven by importation. Interestingly, when we looked at the impact of travel within urban and rural stratum, the difference in prevalence in 2019 between high and low travel areas, as well as the change in prevalence observed overtime and comparing groups, was greater in the rural strata (−10.4%, 95% CI: −19.5, −1.2) compared to the urban strata (−8.3%, 95%CI: −14.8, −1.8). This may be explained by the higher receptivity to onward transmission of imported cases in rural areas. If this is the case, imported infections would demonstrate a multiplicative effect on prevalence because they contribute to more secondary infections than imported cases in urban areas.

Previous analyses have suggested there may be areas where malaria prevalence is solely attributable to imported infection[7], as several of these locations are in urban centers where there is generally improved infrastructure and fewer mosquitoes. If this were true, it would have been anticipated that the risk in high travel areas would be substantially less relative to low travel areas once importation was eliminated. However, this analysis showed sustained prevalence and no difference in odds of infection in the absence of travel. One explanation for this finding is that there was still residual travel occurring, even with the travel moratorium, which allowed infections to continue to be imported during 2020. While a small percentage of individuals did report travel in the past 8 weeks in the 2020 MIS (1%), it seems unlikely that this would sustain the observed prevalence in the population of high travel areas. Another explanation is that infections in high travel areas were acquired through travel to other areas of the island. There is often frequent travel within the island, especially between Malabo and areas in the periphery, where the force of infection is significantly higher[12,14]. In 2019, residents in both high and low travel areas reported an average of 1 trip to another part of the island in the past 8 weeks (range: 1-5 trips), which did not substantially change in 2020. Therefore, it is possible that the remaining prevalence in high travel areas is from within-island parasite movement. However, there was no significant difference seen between prevalence in those who reported within island travel and those who did not in 2020 (Supplementary Table 9), suggesting this does not offer a full explanation. A final possibility is that high travel areas are receptive to local transmission, and levels of endemic transmission persist even when imported infections are removed. This is supported by a 2019 incidence study conducted in Malabo, which suggests local transmission is occurring in peri-urban areas in Malabo district[20]. In that study, while travelers tended to be more likely to have an infection, no incident infections related to travel were identified, supporting the hypothesis that there is local transmission occurring, even in areas where travel is common. In addition, recent entomological monitoring in urban Malabo using human landing catches[21] and larval collections have confirmed the presence of anopheline vectors showing varying levels of human biting rates and larval densities across the city (Supplementary Fig. 2). Therefore, the results of this analysis suggest that control strategies that aim to reduce the malaria burden in travelers, either by reducing the burden in the areas where they travel, and/or by treating returning travelers, would impact the overall prevalence in several communities in Bioko. Additionally, control measures that aim to reduce local transmission, such as IRS, distribution of LLINs, and larval source management should be continued, even if additional interventions that target imported infections are introduced. Further analyses are needed to better understand the role of importation and local transmission at a more granular level.

The interpretation of our results depends on several assumptions. First, it is assumed that the change in prevalence in low travel areas is a valid estimate of the change we would expect in the high travel areas in the absence of imported infections. That is, the two areas had parallel trends prior to the elimination of travel[19]. While it is difficult to definitively verify this assumption, comparing data in high and low travel areas from 2015-2019 suggests similar trends over time. In 2017, the difference in prevalence between the two groups was significantly smaller than in all other years, possibly due to some large, transient increases in prevalence in a few low travel areas due to construction projects. Changes in the ecological landscape are difficult to measure and may have also differentially impacted low and high travel areas during our analytic period. For example, an outbreak occurred in 2019 in a low travel area in the south of the Island, because of recent construction that created additional breeding sites[22]. While there was no precise measure of land use changes to allow for direct adjustment in the model, we did have anecdotal information indicating seven EAs known to have major changes in land use, three of which were included in our analysis. When these EAs were removed from the analysis, the difference in differences and ratio of ratios were slightly attenuated, but still of similar magnitude and significance. Therefore, if it were possible to precisely measure changes in the ecological landscape and include them in the model, we may expect a slightly lower prevalence difference, but the conclusions would likely remain the same. Another variable that was not available at a granular level was rainfall, which can impact mosquito abundance. Inconsistent secular trends are problematic in DID models if they differentially impact the areas of interest. While there have been changes in the monthly amount of rainfall over time on Bioko, there have been increases both in high and low travel areas, and the assumption is that this would equally impact malaria transmission potential in these areas. Finally, our model assumes that enumeration areas were correctly classified as low and high travel areas. While we have high confidence in the assignment of households to the correct EA because the survey is managed through a spatial decision support program[18], there are some low travel areas that are situated immediately next to a high travel area. Given that mosquitoes could travel between these two areas, there could be spillover effects from the imported cases into neighboring areas. If that were the case, we would expect that the prevalence in these neighboring EAs would also have decreased in 2020 compared to 2019. That was not observed, and the mean prevalence in these EAs increased 5% between the years, consistent with the overall observed trends. Calculation of travel prevalence also does not account for the frequency of trips nor possible malaria prevention measures taken while in the higher burden area, which can impact the probability of acquiring an imported infection. Having more detailed information on infection risk while traveling or detailed molecular information to classify imported and locally acquired infections would have allowed us to better delineate areas with high levels of imported cases.

Despite minimal disruptions to the distribution of antimalarials during the study period, we did note that limited MIS data suggested that care seeking decreased by about 10% in both arms between 2019 and 2020, which is similar to data reported through the DHIS2 surveillance system during this time period (internal communication with G.A. García). This decrease in care-seeking could result in increased prevalence, which may be more profound in high travel areas where there was less care-seeking overall. However, adjusting for a community-level care-seeking variable in our model did not impact the results. This is most likely because the decrease was similar in both arms, and not large enough to heavily influence prevalence.

The emergence of SARS-CoV-2 in early 2020 disrupted health systems around the world. As countries closed borders, limited movement, and restricted activities to curtail the initial spread of COVID-19, other public health programs were impacted. This is especially true for many malaria-endemic countries, in which COVID-19

restrictions and global supply chain issues resulted in disruptions in the distribution of long-lasting insecticide nets, application of insecticides, and availability of anti-malarial medicines[15,23,24]. The World Health Organization modeled the potential impact of disruptions to malaria interventions and estimated these disruptions could increase cases by upwards of 20% and deaths by greater than 50%, especially in scenarios where access to treatment was disrupted[25]. Similar impacts were seen during the Ebola outbreak in 2014-2015 when health systems were disrupted[26–29]. However, in these models and analyses, the potential impact of reducing importation and movement of *Plasmodium* infections was not considered[30,31]. This analysis shows that on Bioko Island, where malaria control interventions remained largely uninterrupted during the pandemic[15], travel restrictions resulted in a decrease in malaria prevalence in areas with a high prevalence of travelers. It is possible that other areas with high proportions of imported infections may also have seen these decreases because of the travel restrictions, despite other interruptions to the health care system. This analysis suggests that the impact of COVID-19 on malaria burden may be underestimated in areas with a high prevalence of travelers. Additionally, as borders are now open and imported infections return, malaria control strategy discussions should include interventions that target these infections to reduce the burden.

## Methods

### Malaria indicator survey structure

The MIS is carried out annually on Bioko Island between August and September, as has been previously described[16,17]. Briefly, information on malaria risk factors, including off-island travel in the previous eight weeks, is collected from selected households. Sampling units are geographically defined enumeration areas (EAs); under this scheme all households were eligible for selection into the survey through a stratified, single-cluster survey design. To guide and track programmatic malaria activities, Bioko Island has been divided into geographically defined map areas which are 1 km x 1 km squares[32]. The 209 map areas that included households were used to define EAs for the MIS. If a map area had at least 100 households, it was its own EA; if there were fewer than 100 households in the map area, several map areas were combined (based on geographical proximity) to create an EA with at least 100 households. Before sampling, the EAs were then divided into two strata based on population density and estimated local residual transmission (LRT), which is the predicted amount of infections acquired locally[7]. To select the sample for the MIS, within each EA, a simple random sample of households was taken using specified sampling fractions for each stratum: 24% for stratum 1 (areas with lower population density and higher LRT), and 4.8% in stratum 2 (areas with higher population density and lower LRT).

All adults provided written consent for testing, and the head of household consented for anyone under the age of 18. All consenting individuals who lived in a selected household and were present during the time of the survey were tested for *Plasmodium* malaria parasites using a CareStart Malaria HRP2/pLDH rapid diagnostic test (RDT) (Access Bio, Somerset, NJ, USA). Individuals who were positive for malaria by RDT were provided with artemisinin-combination therapy (ACT) by a Ministry of Health and Social Welfare (MoHSW) nurse per national policy, based on World Health Organization guidelines[33]. Data from the MIS is collected electronically through an ODKCollect form managed on an in-house Android application.

### Sample selection for analysis

Smoothed mainland travel prevalence (the fraction of people surveyed who reported having traveled to the mainland in the eight weeks prior to the survey) for each map area was estimated using travel data from the 2015 to 2018 MIS, using the R-INLA package as per methods described in Guerra et al.[7]. For map areas with no estimates, the value from their nearest neighbor was utilized. If an EA was composed of multiple map areas, a weighted average was calculated from all map areas in the EA. The weight of each map area was equal to the number of households in that area out of the total number of households in the EA.

After a historical travel prevalence was assigned to each EA, those in the top quartile of travel prevalence were labeled as high travel areas, and those in the bottom quartile of travel prevalence were labeled as low travel areas; EAs from the middle two quartiles were excluded from analysis (Fig. 3).

### Statistical analyses

All statistics were performed using R statistical software (v3.6.2). Analyses of survey data were conducted within the *survey* package (v4.1-1). The survey design dataset accounted for the stratified sampling weights of the original MIS as well as the non-independence of results within households and within EA. The main outcome of interest, *Pf* positivity, was coded as a binary variable. For each travel area, the survey mean prevalence was estimated from individual level data by year and are presented with a 95% CI.

To analyze the possible impact of the travel moratorium on malaria risk on Bioko Island, a difference in differences analysis was conducted to compare the difference in prevalence of infection between 2019 and 2020 in historically high travel areas relative to the difference in prevalence in historically low travel areas during the same time. For our main analysis, we fit an unadjusted and adjusted survey generalized linear model with robust standard errors. To determine variables to include in the adjusted model, we compared values of several variables determined to be related to malaria risk a priori between 2019 and 2020 within travel group. Any variable that had a meaningful difference between years within a travel group was included in the final model. A meaningful difference was defined as an absolute change of at least 5% and proportional change of at least 10%. Analysis was done with observations that had complete data for all variables in the fully adjusted model. To estimate how the prevalence of infection in high travel areas changed between 2019 and 2020 relative to the change in prevalence in low travel areas over the same period, the model included an interaction term between a binary variable for time and travel. Coefficients and 95% CIs were extracted for various combinations. The general model is presented in Eq. 1:

$$Pr(RDT+) = \beta_0 + \beta_1 POST + \beta_2 hightravel + \beta_3 POST * hightravel + \beta_4 Covars + \varepsilon \tag{1}$$

Where $\beta_0$ is prevalence of malaria infection in low travel areas in 2019, $\beta_1$ is the difference in prevalence comparing 2020 to 2019 in low travel areas, $\beta_2$ is the difference in prevalence of infection between low travel and high travel areas in 2019, and $\beta_3$ is the difference in differences of prevalence comparing the change in high travel areas between 2019 to 2020 to the change in low travel areas from 2019 to 2020. $\beta_4$ represents a vector given the covariates in matrix Covars that represent the various covariates included in respective models, and $\varepsilon$ is the residual variance. $\beta_3$ is the coefficient of interest to estimate the impact of the travel restrictions on prevalence.

Given that our outcome was binary, we also evaluated the relationship between odds of infection in high and low travel groups between years using logistic regression. The same models were fit but utilizing a logit link function. Coefficients and 95% CIs were exponentiated to get comparative odds ratios between years and travel areas.

Data from the 2015–2018 MIS was used to assess the robustness of the parallel trends' assumption by visually assessing the trends from 2015 to 2019 in high and low travel areas and fitting a linear mixed effects model with an interaction term for each year and travel stratum in the pre-moratorium data[34]. To test the robustness of the model, we performed a bootstrapped analysis using a randomly selected 80%

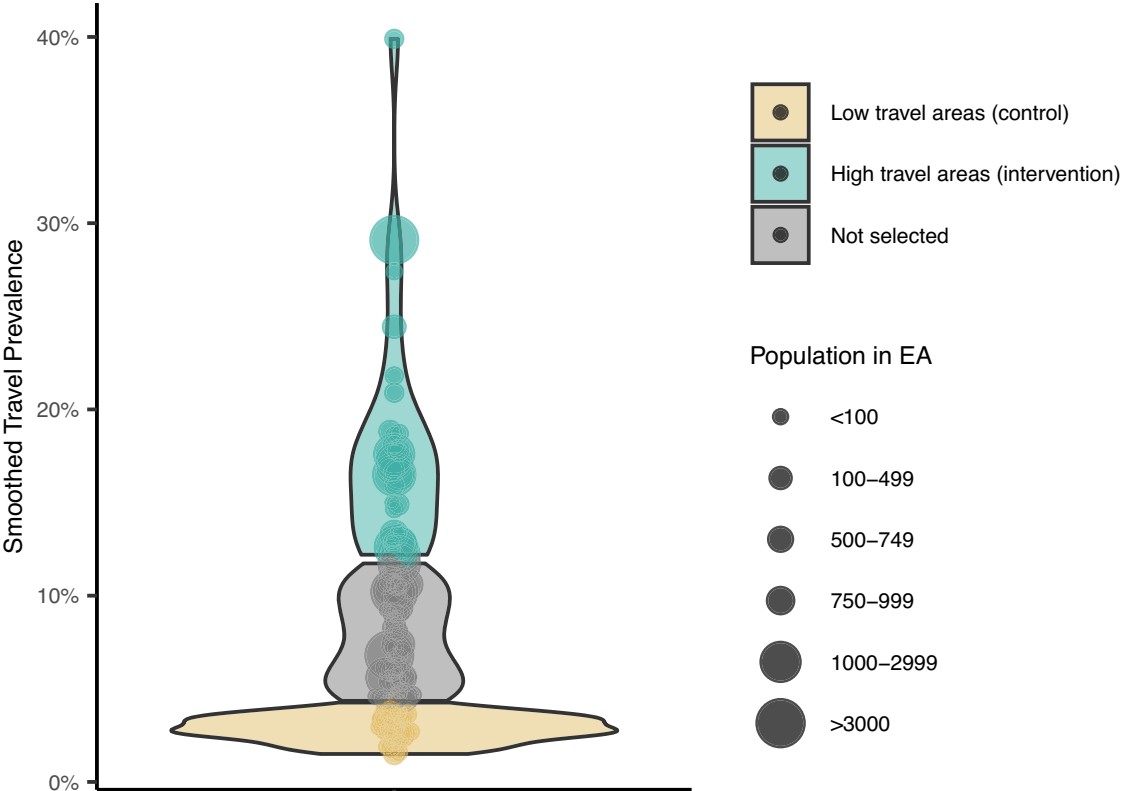

**Fig. 3 | Distribution of travel prevalence by Enumeration Area (EA).** Violin plot showing the distribution of travel by EA for all EAs, color coded by travel area classification, based on smoothed travel prevalence estimates from 2015-2018[7]. EAs in the top quartile of travel distribution (teal) were classified as high travel areas, and those in the bottom quartile of travel distribution (gold) were classified as low travel areas. Areas in the middle two quartiles (gray) were not included in the analysis.

sample of the data, repeated for 50 iterations. A count of the number of times each interaction term was significant (suggesting non-parallel trends) was calculated. For the analysis of parallel trends, non-survey weighted prevalence was calculated each year, as sample selection in 2015–2018 was not done in the same manner as subsequent years.

There were seven EAs known to have had large land use changes over the study period, three of which were in our analytic dataset. As there was not a reliable way to measure land use change in all areas during the study period, we conducted a sensitivity analysis, in which the main analysis was repeated with a data set that excluded the three EAs that were known to have had land use changes over the study period.

### Ethics & inclusion statement

Approval for the annual implementation of the Malaria Indicator Survey was provided by the Ministry of Health and Social Welfare (MoHSW) of Equatorial Guinea. This analysis was determined to be exempt from further IRB review by the University of Washington Human Subjects Division (STUDY00012460).

The National Malaria Control Program approved the exploration of this research question. Development of the analysis plan, review of possible confounders, and discussion of preliminary and final results was done by a collaboration of local and global researchers. Understanding of locally relevant information, such as land use changes and programmatic challenges during the study period was instrumental in the correct interpretation of results.

### Reporting summary

Further information on research design is available in the Nature Portfolio Reporting Summary linked to this article.

### Data availability

The data that support the main findings of this study are available in the GitHub repository [https://github.com/d-hergott/DID-public][35].

### Code availability

Reproducible code for the main analysis is available in the GitHub repository [https://github.com/d-hergott/DID-public][35].

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

## Acknowledgements

We thank the participants and survey teams of Bioko Island who participated in the Malaria Indicator Survey. Malaria control on Bioko Island, including the Malaria Indicator Survey, is funded by the Government of Equatorial Guinea in partnership with a consortium of private companies led by Marathon Oil Corporation. Support for this work was also provided by grants from the National Institutes of Health (R01 AI163398, DLS) and the Bill and Melinda Gates Foundation (INV 030600, DLS).

## Author contributions

D.H. and J.B. conceptualized the analysis. D.H., J.B., J.N.M.E., M.M.I., G.A.G., and O.D. designed the analytical plan. J.N.M.E., O.D., M.M.I., R.M.N.A., C.N.A.N., and T.A.O.M. collected and cleaned the survey data. M.R.R. and W.P.P. provided operational oversight and support. C.A.G. calculated travel prevalences. D.H. carried out the analyses, created the figures, and drafted the manuscript. C.A.G., G.A.G., S.C.M., D.L.S., B.G., and J.B. provided critical review and editing of the manuscript. All authors reviewed and approved the final manuscript.

## Competing interests

The authors declare no competing interests.
