## [Peer Review File · Nature Communications]

Impact of six-month COVID-19 travel moratorium on *Plasmodium falciparum* prevalence on Bioko Island, Equatorial GuineaREVIEWER COMMENTS

Reviewer #1 (Remarks to the Author):

Overall the manuscript is well written, the methods are clear and reasonable, and the conclusions are interesting and provide insight into the hard-to-parameterize nature of malaria importation.

Strengths - see above

Weaknesses - the most concerning issue I see is the failure to include changes in care-seeking during 2020 as a variable within the analysis. This is somewhat understandable given it would add the complexity of deriving and then including a community level variable in the model. However, the WHO Pulse surveys suggest that Equatorial Guinea had one of the higher rates of healthcare disruption in SSA, and it would be unsurprising if this disruption were asymmetrical (e.g., more severe in rural EA that also tend to be "low travel" areas). Furthermore, if we're meant to assume that travelers returning with plasmodium increase the odds that others living in their EA will be infected, surely a changing proportion of individuals in the community clearing their infections with an antimalarial drug needs to be considered too. While I couldn't find a published report for the 2020 survey, the 2019 report showed that the standard MIS data for care seeking for fever were collected. I strongly suspect that these data were also collected in 2020. As such, I suggest the authors redo the analysis to include treatment seeking differences among the high and low travel EAs. If not, they need to provide a strong justification for why this variable was omitted and at least discuss how it might have influenced their results.

I'm also curious why ITN use wasn't adjusted for in the analysis. Decreases in ITN were greater in low travel (6.1%) than in high travel (3.7%), and these changes were of similar magnitude to the "go inside before 7" metric that was adjusted for.

Minor suggestions - (i.e., possibly worth a clarifying sentence)

1) The EA areas (oddly represented as mostly square blocks, which is consistent with the survey report) are fairly small. Estimating via the scale bar on the map, they look to be around 2 to 2.5 km in size, which is considerably smaller than the typical level of displacement applied by DHS when conducting surveys. There are several instances of adjacent high- and low-travel EAs. Any concern about household clusters being misattributed? If not, why not?

2) text edit to line 204 - "high travelers". Potentially replace with "frequent travelers" or "a high prevalence of travelers"

3) The legend for Figure 2 is unclear although still somewhat intuitive. The hypothetical (dashed) line needs a legend symbol that differentiates it from the observed line.

Reviewer #2 (Remarks to the Author):

This paper is an interesting approach to considering how imported cases affect transmission trends on Bioko Island. The research makes use of a natural enforced break in travel given COVID-19 restrictions. The conclusions are sensible given the methods, and the draft is well presented. We are generally supportive of its publication and conclude that it fits the journal reasonably. However, we note below some concerns and make some suggestions that we think need to be addressed prior to acceptance.

Major considerations

Given that much of the 'high travel' regions are also more urban, we are concerned that trends observed are potentially more reflective of the urban-rural differences that exist for malaria

transmission. Would the authors consider looking at a sub-set of sites that could be characterised as 'high travel' and rural and 'low travel' and urban spaces? If the patterns hold here, the inference about importation could be further supported. Population density relative to mosquito density is a major part of this given urban settings tend to have fewer bites per person than rural settings. Vectorial Capacity says that the ratio of people to mosquitoes is critical – so if more people in high travel zones, and less mosquitoes – this would suggest lower prevalence across the population would be expected compared to less densely populated, higher mosquito zones that sound like they are more descriptive of the low travel places. Could the authors comment on whether the higher burden in the high travel sites in 2018 and 2019 relative to low travel sites may be principally due to importation, or whether any other factors can explain this?

It was interesting to see the increased IRS and the earlier time to go indoors indicated by the data – both would theoretically reduce exposure to infectious bites. We suggest discussing the earlier indoor action in 2020 in more detail perhaps in relation to how the pandemic may have indirect impacts through altered behaviours.

The methods note that you have asked 'did the person travel in the past n weeks? With a yes or no response. We wondered how frequency of visits would alter the conclusions. Could you add a sentence or two to the discussion to acknowledge this limitation. We also wondered if you had considered within and between travel i.e. there are a few sites on the map, within the urban centre that are characterised as low travel but, these people may well move regularly within the sites noted as high travel in the same urban space. Could you comment on any impact this might have and whether you considered ways to explore onward transmission?

The analysis is reasonable except for the trends analysis where the authors consider only the 2018 to 2019 as a 'ground-truth' of the changing trend due to the reduced travel. We suggest that you consider a few more year-on-year differences. Can you look at the past 5 years for example and show less year-on-year change than that observed in 2019 to 2020... Can you comment on whether, in the past 5 or 10 years, there have been any other year-on-year changes of the same magnitude as that reported for 2019 to 2020? If so, are there similarly events that could explain these? Any other travel reductions/increases, any major discrepancies in local weather for the respective groups, etc? This would help consolidate that the observation is indeed driven by the reduced travel.

Can you also show some uncertainty on the trends analysis. We suggest that you try bootstrapping the data, repeat your analysis with 80% of the samples say 50 times and see how many times you generate the same conclusion? This would give some confidence to the findings perhaps.

Throughout the methods, we think the contribution would benefit greatly from writing out the models fitted more fully. Perhaps providing code via github or another platform, and at least noting the model equations in the supplement.

Did the authors consider different Plasmodia? In PNG/Indonesia, near elimination settings have a skewed ratio toward more vivax than falciparum infections – does the MIS data report the species of infections? Could the authors comment on this?

Can you comment on any reasons why the increase in prevalence occurred in 2020 for low travel zones? Was there issues for drug treatment getting to people due to Covid? Any increased rainfall? Were these driven by the land use change locations discussed? (This links with the note above discussing the potential induced behaviour change observed as a higher percentage of people going indoors before 7 pm).

Minor considerations

Define EAs line 47 (it is defined in the Figure but not previously)

Given the journal structure, we suggest defining "travel prevalence" in the results section because it is otherwise unclear what is meant by the term. (It is clearly defined in the methods)

Please consider whether any weighting is needed in the model to accommodate population sizes?

Line 187, suggest adding "... the conclusions would likely remain the same." As without performing the analysis, it is not reasonable to say this.

Thank you for the opportunity to review.

Reviewer #3 (Remarks to the Author):

Response to Reviewers

Title: Impact of six-month COVID-19 travel moratorium on *Plasmodium falciparum* prevalence on Bioko Island, Equatorial Guinea

We thank the reviewers for their insightful and thought-provoking questions. Updates have been made to the paper in response to these comments, and in line responses are provided below.

Additionally, we want to note that when making the updates suggested by reviewers, we noted two errors in our original analysis. The first is that the “go indoors before 7pm” variable was incorrectly coded for 2020 survey data in such that individuals who reported going in between 7PM and 8PM were classified as “going in before 7pm”. Correction of this coding resulted in much less drastic changes between years for this variable, but the differences were still large enough that we included the variable in our final model.

Secondly, there were some enumeration areas that had not received IRS during the study years. Instead of being coded as 0% coverage, these areas were coded as NA, and were inadvertently excluded from the model.

Correction of these two variables resulted in an adjusted DID value that was greater than the unadjusted variable. **This does not change the results or the conclusions of the paper, but simply points to a possible larger impact of importation.**

Reviewer #1 (Remarks to the Author):

Overall the manuscript is well written, the methods are clear and reasonable, and the conclusions are interesting and provide insight into the hard-to-parameterize nature of malaria importation.

Strengths - see above

Weaknesses - the most concerning issue I see is the failure to include changes in care-seeking during 2020 as a variable within the analysis. This is somewhat understandable given it would add the complexity of deriving and then including a community level variable in the model. However, the WHO Pulse surveys suggest that Equatorial Guinea had one of the higher rates of healthcare disruption in SSA, and it would be unsurprising if this disruption were asymmetrical (e.g., more severe in rural EA that also tend to be “low travel” areas). Furthermore, if we’re meant to assume that travelers returning with plasmodium increase the odds that others living in their EA will be infected, surely a changing proportion of individuals in the community clearing their infections with an antimalarial drug needs to be considered too. While I couldn’t find a published report for the 2020 survey, the 2019 report showed that the standard MIS data for care seeking for fever were collected. I strongly suspect that these data were also collected in 2020. As such, I suggest the authors redo the analysis to include treatment seeking differences among the high and low travel EAs. If not, they need to provide a strong justification for why this variable was omitted and at least discuss how it might have influenced their results.

This is an excellent point, and a variable that we did not initially consider, because of the small number of individuals (~10%) who were sick in the two weeks prior to the survey, and therefore provided an answer about care seeking behavior. However, as the reviewer suspected, it did appear that fewer people sought treatment in *both* arms in 2020 compared to 2019. While the change in care seeking was mostly symmetrical, given that the baseline values in each were different, we chose to evaluate this as a possible adjustment variable in a sensitivity analysis. We created a community level variable for care seeking using the survey responses and included it in the model. There was no impact on the estimates when this variable was included in the model. It is possible we did not see an impact because of the high variability of this response. For example, some EAs had as few as 4 individuals who reported being sick in the past two weeks, meaning that the community level care seeking variables could be highly influenced by the behaviors of one or two individuals that are not necessarily reflective of the community.

I'm also curious why ITN use wasn't adjusted for in the analysis. Decreases in ITN were greater in low travel (6.1%) than in high travel (3.7%), and these changes were of similar magnitude to the "go inside before 7" metric that was adjusted for.

Bednet ownership by household decreased slightly in low travel areas (46.2% vs. 43.9%) while increasing in high travel areas (26.1% vs. 34.6%). However, the proportion of survey respondents who indicated sleeping under a bednet the previous night was very similar in 2019 and 2020 in low travel (44.6% vs. 41.2%) and high travel (36.3% vs. 34.9%). As bednet use is more predictive of protection than household ownership, we did not adjust for this metric.

Minor suggestions - (i.e., possibly worth a clarifying sentence)

1) The EA areas (oddly represented as mostly square blocks, which is consistent with the survey report) are fairly small. Estimating via the scale bar on the map, they look to be around 2 to 2.5 km in size, which is considerably smaller than the typical level of displacement applied by DHS when conducting surveys. There are several instances of adjacent high- and low-travel EAs. Any concern about household clusters being misattributed? If not, why not? **The data from this survey is not collected as part of DHS. The survey was adapted from the DHS survey and is administered by the program. The Program uses a spatial decision support system that is robust, error-controlled, and geographically accurate. The EAs are mostly square blocks because the program operates in sector and map areas, which are based on a grid. As such, all clusters are correctly attributed to the correct EA.**

Given that, we do recognize that they are close together, and added the possibility for misclassification or spillover effects as a limitation in our discussion.

2) text edit to line 204 - "high travelers". Potentially replace with "frequent travelers" or "a high prevalence of travelers"

This has been updated with the suggested text.

3) The legend for Figure 2 is unclear although still somewhat intuitive. The hypothetical (dashed) line needs a legend symbol that differentiates it from the observed line.

The legend symbol has been updated to show the dashed line.

Reviewer #2 (Remarks to the Author):

This paper is an interesting approach to considering how imported cases affect transmission trends on Bioko Island. The research makes use of a natural enforced break in travel given COVID-19 restrictions. The conclusions are sensible given the methods, and the draft is well presented. We are generally supportive of its publication and conclude that it fits the journal reasonably. However, we note below some concerns and make some suggestions that we think need to be addressed prior to acceptance.

Major considerations

Given that much of the 'high travel' regions are also more urban, we are concerned that trends observed are potentially more reflective of the urban-rural differences that exist for malaria transmission. Would the authors consider looking at a sub-set of sites that could be characterised as 'high travel' and rural and 'low travel' and urban spaces? If the patterns hold here, the inference about importation could be further supported. Population density relative to mosquito density is a major part of this given urban settings tend to have fewer bites per person than rural settings. Vectorial Capacity says that the ratio of people to mosquitoes is critical – so if more people in high travel zones, and less mosquitoes – this would suggest lower prevalence across the population would be expected compared to less densely populated, higher mosquito zones that sound like they are more descriptive of the low travel places. Could the authors comment on whether the higher burden in the high travel sites in 2018 and 2019 relative to low travel sites may be principally due to importation, or whether any other factors can explain this?

We appreciate this comment, and it is true that most of the higher travel regions are more urban than lower travel regions. Given that, as the reviewers note, urban areas tend to have more people and fewer mosquitoes, we would expect the prevalence to be lower in these areas than in more rural areas with greater mosquito density. The fact that this is not true in Bioko and urban areas have had persistent higher than expected prevalence is what initially lead to investigations into the impact of importation to the possible prevalence in these areas.

Stratum, which is included in our survey model to account for sampling weights, serves as a proxy for urban/rural. Stratum 1 represents rural areas (lower population density and higher local residual transmission) while stratum 2 represents urban areas (higher population density and lower local residual transmission). Low travel areas were made up of 61% rural areas (stratum 1), while high travel areas were made of 82% urban areas (stratum 2). Therefore, as noted by the reviewer, high and low areas do somewhat represent urban and rural division. However, the only major change in urban areas between 2019 and 2020 that could account for the dramatic difference in prevalence

was the stoppage of travel to the mainland, and presumably the limit of importation of cases to these areas. Most of the land use changes, which have led to outbreaks in the past, were in rural areas, and we were able to adjust for these in our sensitivity analysis.

Other factors, including changes in spray coverage, more houses with air conditioning units, and within island travel, which may explain some of the differences in prevalence between years were also included in our model, so our effect estimate is what we believe to be the impact of importation after accounting for other explanatory factors. We have hypothesized that the higher burden of malaria in high travel sites in previous years (now shown back to 2015) is principally due to importation, and we believe this current study supports that claim.

Per the reviewer’s suggestion, we did also run the analysis stratified by urban/rural (using stratum), as well as among the subset of EAs that were classified as rural high travel and urban low travel. In the stratified analyses (model outputs shown below), we can see that the difference in prevalence between high and low travel areas in 2019 (*trav var*) is greater in rural areas (12%) than in urban areas (8%). This is most likely because there is greater susceptibility to onward transmission of imported infections in rural areas, so the impact of importation is magnified. Similarly, the *DiD* term, or difference in prevalence between 2020 and 2019 comparing high areas to low areas (*trav.year.b* term) was 10% in rural areas and 8% in urban areas. Similarly, this suggests that removing importation in rural areas, which are more susceptible to onward transmission, had a slightly greater impact than removing imported cases in rural areas.

	Stratum 1/Rural n= 3103				Stratum 2/Urban n=9025			
	est	lcl	ucl	p-val	est	lcl	ucl	p-val
X.Intercept.	0.057	0.019	0.094	0.006	0.065	0.027	0.104	0.002
trav	0.123	0.023	0.224	0.019	0.078	0.042	0.115	0.000
year.b	0.062	-0.003	0.128	0.061	0.055	-0.009	0.119	0.089
inbefore7	-0.032	-0.073	0.008	0.106	-0.031	-0.047	-0.016	0.000
spry_perc	0.107	0.015	0.200	0.026	0.041	-0.004	0.086	0.074
aircon	0.022	-0.114	0.157	0.737	-0.029	-0.054	-0.004	0.023
travelledisland	0.000	-0.028	0.029	0.998	0.010	-0.017	0.037	0.445
trav.year.b	-0.104	-0.195	-0.012	0.029	-0.083	-0.148	-0.018	0.014

We then also evaluated the relationship, limiting the data to only those low travel areas from urban areas (11 EAs) and high travel areas from rural stratum (5 EAs). In this model, prevalence in the high travel areas decreased 6.9% more than what was expected (95%CI: -15%, 1.5) in the unadjusted model and 8.5% more than expected in the fully adjusted model (95% CI: -18.05, 1.9%). The result was bordering on significance, most likely due to the vastly reduced sampled sized (n=2542).

We hope that these additional analyses ease concern that the impact we are seeing is due to something other than importation.

It was interesting to see the increased IRS and the earlier time to go indoors indicated by the data – both would theoretically reduce exposure to infectious bites. We suggest discussing the earlier indoor action in 2020 in more detail perhaps in relation to how the pandemic may have indirect impacts through altered behaviours.

Thank you. As noted above, we noticed a small error in our coding that erroneously adjusted our DiD in the opposite direction in our initial analysis. Now, earlier time indoors did indeed show reduced malaria prevalence, with prevalence in those who went indoors before 7PM 3% less than those who did not.

The IRS result is a nuisance of the targeted spray approach. IRS is applied only to areas with the highest malaria burden. As such, having high IRS coverage is associated with higher burden. This is most likely why increased coverage is associated with increased malaria prevalence in the model, even though we would expect IRS to decrease prevalence.

The methods note that you have asked ‘did the person travel in the past n weeks? With a yes or no response. We wondered how frequency of visits would alter the conclusions. Could you add a sentence or two to the discussion to acknowledge this limitation.

.

Without more detailed information, it is difficult to hypothesize the impact of more frequent visits would have on importation risk. We would assume that the more trips an individual takes, the higher their risk of acquiring an infection would be. However, risk is also dependent on mosquito bite prevention behaviors that the traveler takes while in the higher prevalence area, such as sleeping under a net, or staying in a house or hotel with closed eaves and air conditioning. The data suggests that the more frequent travelers are often members of the government (who must go back and forth for official work purposes), and, anecdotally, they stay in higher quality housing while on the mainland. Therefore, while their risk is increased because of more frequent trips, it is simultaneously decreased by these prevention behaviors. This level of detail is not collected during the survey, so these nuisances cannot be further explored. If we had information gathered in a different way, that both accounted for frequency of travel as well as possible malaria exposure while travelling, we could have better delineated areas with high importation to those of low importation. Unfortunately, we did not have that data.

Frequency of travel (number of trips taken in the past 8 weeks) is asked about in the survey and was a covariate evaluated in our supplementary table one and mentioned in the discussion. Not surprisingly, those in high travel areas take more trips, on average, than those in low travel areas. In 2019, most individuals who reported travel in the past 8 weeks took one trip in both high travel (232/369) and low travel (32/40) areas. About a quarter of travelers in high travel areas took 2 trips (105/369).

We also wondered if you had considered within and between travel i.e. there are a few sites on the map, within the urban centre that are characterised as low travel but, these people may well move

regularly within the sites noted as high travel in the same urban space. Could you comment on any impact this might have and whether you considered ways to explore onward transmission?

Thank you for this comment. The MIS does include a variable for within island travel (for those who spent at least one night in a community that was not theirs). The possible impact of this was presented in the discussion (lines 186-194). Additionally, we added in within island travel to our model, as there was a decrease in this type of travel in both arms, and was imbalanced at baseline between arms. However, this did not alter the results. We have added some additional information about possible misclassification and onward transmission to the discussion.

The analysis is reasonable except for the trends analysis where the authors consider only the 2018 to 2019 as a 'ground-truth' of the changing trend due to the reduced travel. We suggest that you consider a few more year-on-year differences. Can you look at the past 5 years for example and show less year-on-year change than that observed in 2019 to 2020... Can you comment on whether, in the past 5 or 10 years, there have been any other year-on-year changes of the same magnitude as that reported for 2019 to 2020? If so, are there similarly events that could explain these? Any other travel reductions/increases, any major discrepancies in local weather for the respective groups, etc? This would help consolidate that the observation is indeed driven by the reduced travel. Can you also show some uncertainty on the trends analysis. We suggest that you try bootstrapping the data, repeat your analysis with 80% of the samples say 50 times and see how many times you generate the same conclusion? This would give some confidence to the findings perhaps.

Prior to 2015, MIS sampling was done differently, with only households from select sentinel sites included- so we were only able to extend this analysis back to 2015. Between 2015-2018, the sampling strategy was different from 2018 onward, but we still had data from all of the high and low travel EAs to conduct the trend analysis with more data. The figure has now been updated to include data from 2015-2020.

Parallel trends appeared to hold in all years with the exception of 2017, when there was a larger increase in prevalence in low travel areas than in high travel areas. One possible reason for this is that in 2017, there were a few urban development projects in some of the low-travel areas that created temporary breeding sites and resulted in increased cases in these areas. When these EAs were removed from the analysis, the increase in low-travel areas was less pronounced and more similar to that in high travel areas.

We added confidence intervals from the modelled prevalence estimates to the graph. We also bootstrapped as the reviewer suggested, and calculated the number of iterations in which the interaction terms of our model for year*travel_group were significant (indicating non-parallel trends). This information has been added into the manuscript.

Throughout the methods, we think the contribution would benefit greatly from writing out the models fitted more fully. Perhaps providing code via github or another platform, and at least noting the model equations in the supplement.

We added the main model into the methods section. We will also include code and blinded data via github to allow users to replicate our process.

Did the authors consider different Plasmodia? In PNG/Indonesia, near elimination settings have a skewed ratio toward more vivax than falciparum infections – does the MIS data report the species of infections? Could the authors comment on this?

The RDTs used for the MIS detect *falciparum* and *non-falciparum* species only. However, the majority of the infections (95%) are either purely *falciparum* or *falciparum* mixed with another infection. *Ovalae* and *malariae* are more common than *vivax* infections in this area

Can you comment on any reasons why the increase in prevalence occurred in 2020 for low travel zones? Was there issues for drug treatment getting to people due to Covid? Any increased rainfall? Were these driven by the land use change locations discussed? (This links with the note above discussing the potential induced behaviour change observed as a higher percentage of people going indoors before 7 pm).

It is not fully known what caused the increase in prevalence in low travel areas for this year. There was an increase in prevalence between 2018 to 2019, so this could be a continuation of that trend and a reflection of current transmission dynamics on the island. Bioko Island historically had one of the highest transmission rates in the world, and the environment is conducive to year-round mosquito breeding and transmission. Prevalence has mainly plateaued in the past 10 years, showing yearly oscillations around the mean.

Based on historical trends, it appears prevalence also increased between 2018 and 2019 in low travel zones. Land use change most certainly has an impact, as evidenced by our analysis, and previous analyses that linked outbreaks with construction, which create many temporary breeding sites that lead to increases in cases. However, removing those areas with known land use changes did not remove the effect we saw.

Covid impacted care-seeking, as we now show in the paper, but the magnitude in the decrease was similar in high and low travel areas and there were no large stock outs or similar issues due to Covid-19 that would explain the increase only in these low travel areas. All facilities were operational and Program continued to provide diagnostics and antimalarials to all health facilities on the island. Following 2020, care seeking in public health data has remained similar to 2021 and 2022, and trends are similar in urban and rural areas.

Vector control was the main intervention during 2020, and there was no larval source management or new distribution of LLINS (other than through prenatal clinics). There were ~10,000 fewer households sprayed in 2020 compared to 2019, and spray teams put less effort into mop-up (going back to households who refused or were absent during the first visit) because of COVID concerns. Our data does show a decrease in spray coverage in low-travel areas, which could lead to increased prevalence. However, including this variable in the model did not greatly impact the change in prevalence in low travel areas from 2019 to 2020, so it does not fully explain the increase.

Unfortunately, rainfall data is only available island wide, and we do not have more detailed information on possible environmental differences between areas of the island. However,

anecdotal reports do not suggest that there were differences between the areas that would've contributed to the increase.

Minor considerations

Define EAs line 47 (it is defined in the Figure but not previously)

Given the journal structure, we suggest defining “travel prevalence” in the results section because it is otherwise unclear what is meant by the term. (It is clearly defined in the methods).

We have added a sentence in the results as suggested.

Please consider whether any weighting is needed in the model to accommodate population sizes?

As complete household lists were not available for each year of the survey, the models were weighted using the sampling weights for each stratum, which accounts for the differences in population sizes in each area.

Line 187, suggest adding “... the conclusions would likely remain the same.” As without performing the analysis, it is not reasonable to say this.

We have made this update as suggested.

Thank you for the opportunity to review.

Reviewer #3 (Remarks to the Author):

Thank you for your detailed review!

REVIEWER COMMENTS

Reviewer #1 (Remarks to the Author):

I am satisfied with the authors' revisions and consider their rebuttals to the points raised in review quite reasonable. Per my initial review, I believe the results of this analysis are noteworthy, novel, and will be of interest to the malaria research community. I have no additional critiques and recommend accepting the revised manuscript for publication.

Reviewer #2 (Remarks to the Author):

Thanks for the revised manuscript submission.

In general, we are supportive of the publication but there remains an important clarification for the methods, and a couple of minor corrections that are necessary, and some suggested actions that the authoring team might consider. Given these relate to the equation and analysis, we have attached thoughts as a separate document so we can lay out our concerns more fully.

Reviewer #3 (Remarks to the Author):

Reviewer #2 (Attachment):

Thanks for the revised manuscript submission.

In general, we are supportive of the publication but there remains an important clarification for the methods, and a couple of minor corrections that are necessary, and some suggested actions that the authoring team might consider.

Most importantly, the equation for the statistical model fitted is confusing.

In the data shared via github, the “trav” and “year.b” covariates are specified. These are binary variables for travel (either NA, 0 or 1 – presumably flagging no data, low or high travel), and year (0 and 1 for 2019 or 2020) respectively. These translate as the *high.travel* and *POST* terms flagged in the model in the manuscript.

In the github code, *falc_pos* is the model outcome (y variable), and is a binary variable showing infection (1) or no infection (0) status. A Gaussian link is assumed, but we think this is inappropriate given the bounded variable (from 0 to 1). Why was the Gaussian link chosen, and what does this add?

The odds ratios are calculated when the function allows the Binomial distribution to be assumed for the data which makes more sense to us. In this case, the reported model in the manuscript should perhaps be:

$$RDT_+ \sim \text{Binomial}(\gamma, RDT_{total})$$
$$\text{logit}(\gamma) = \beta_1 \text{post} + \beta_2 \text{high.travel} + \beta_3 \text{post} \times \text{high.travel} + \beta_4 \mathbf{X} + \varepsilon$$

Here, the probability gamma is estimated assuming the binomial data, given positive (RDT_+) cases out of all (RDT_{total}) tested and reported individuals. The explanations in the text (with the exception of β_2 see below) hold, but here, β_4 is a vector given the covariates in matrix \mathbf{X} that represent the different covariates included in the respective models. The unexplained error is described by (ε). In the edited manuscript neither β_4 or ε are currently explained. Please add this.

The variable “stratum” is a value of 1 or 2 and is also used to make the survey design object in the github code. What is this term in the context of this work? The survey package description notes “For multistage sampling the id argument should specify a formula with the cluster identifiers at each stage. If subsequent stages are stratified “strata” should also be specified as a formula with stratum identifiers at each stage.”

The edited manuscript describes β_2 as the difference in prevalence of infections between low- and high- travel in 2019, but is this not the difference between low and high travel at any time?

Second, it would be very useful to overlay the prevalence points for the MIS figure (Figure 2) for each year. Could you also note whether these surveys are repeated in sentinel villages or are villages randomly chosen to represent an Island average? Then you can comment on any limitations. The uncertainty bars on Figure 2 look fairly consistent so plotting the points on top for the respective surveys could be a good way to visualise how well these represent the MIS data. You could jitter points and stagger them for the green and yellow lines.

Finally, we thank the authors for diligently addressing the points raised during the first review. The manuscript was very clearly written previously and now contains a couple of rushed sections – these are added to address reviewer comments. We suggest tightening writing in these sections to ensure typos are eliminated and points are not repeated. But all necessary details are there, and it is an interesting approach worthy of publication.

Thanks for the revised manuscript submission.

In general, we are supportive of the publication but there remains an important clarification for the methods, and a couple of minor corrections that are necessary, and some suggested actions that the authoring team might consider.

Most importantly, the equation for the statistical model fitted is confusing.

In the data shared via github, the “trav” and “year.b” covariates are specified. These are binary variables for travel (either NA, 0 or 1 – presumably flagging no data, low or high travel), and year (0 and 1 for 2019 or 2020) respectively. These translate as the *high.travel* and *POST* terms flagged in the model in the manuscript.

In the github code, *falc_pos* is the model outcome (y variable), and is a binary variable showing infection (1) or no infection (0) status. A Gaussian link is assumed, but we think this is inappropriate given the bounded variable (from 0 to 1). Why was the Gaussian link chosen, and what does this add?

Your interpretation of the data is correct, *falc_pos* is a binary variable, indicating if an individual was positive for *Pf* via RDT during the time of the survey. Given that it is binary, we originally did our analysis using a logit link to calculate the odds of infection in each study arm, and then calculated odds ratios to show changes between years and arms, and a “ratio of ratios” as our DID term. However, we encountered difficulties as we communicated the results to our collaborators. We found that the comprehension and interpretability of an odds ratio, and particularly the DID term, was difficult and not easily digested by our academic and programmatic partners.

As such, we chose to present data as prevalence differences, which can be estimated using the number of positives over total tests, or the mean of the *falc_pos* variable. This mean is provided using a linear survey model. While a Gaussian link is not the most appropriate statistical model, given that our data was not particularly close to either 0 or 1, and we were not trying to extrapolate any values, we opted to utilize the more interpretable statistical analysis. The modeled prevalences using the Gaussian link (and linear combinations of the outputs) were almost identical to those obtained using *svymean* calculations (without adjustment), suggesting that the model was accurate.

However, knowing that this analysis was not the most statistically appropriate, we also present the results from the logistic model in several places throughout the text and discuss the difference in odds in the discussion. We also had already included the full logistic model analysis in an appendix (Supplementary Table 3 and Supp Figure 1) so readers (and reviewers) can see both results, and understand that the conclusions are the same on a relative and absolute scale.

Below is a side by side of the various comparisons on the absolute and relative scales is presented on the next page.

Unadjusted model results for various comparisons using Gaussian link (linear model) and logit link (binomial).

Comparison-Linear	Linear model PD (95% CI)	Comparison-Relative	Binomial Model OR (95% CI)
High travel areas 2020-High travel 2019	-1.9 (-3.2,-0.5)	High travel areas 2020/High travel 2019	0.84 (0.75,0.96)
Low travel 2020-Low travel 2019	5.5 (0.9,10.1)	Low travel 2020/Low travel 2019	1.86 (1.22,2.84)
High travel 2020-Low travel 2019	4.4 (1.1,7.7)	High travel 2020/Low travel 2019	1.69 (1.08,2.63)
High travel 2020-Low travel 2020	-1.1 (-7,4.8)	High travel 2020/Low travel 2020	0.91 (0.53,1.54)
High travel 2019-Low travel 2019	6.3 (3.3,9.4)	High travel 2019/Low travel 2019	2.0 (1.30,3.05)
High travel areas 2020-high travel areas 2019) - (Low travel areas 2020-low travel areas 2019)	-7.4 (-12.1,-2.6)	High travel areas 2020/high travel areas 2019 to Low travel areas 2020/low travel areas 2019	0.45 (0.29, 0.71)

The odds ratios are calculated when the function allows the Binomial distribution to be assumed for the data which makes more sense to us. In this case, the reported model in the manuscript should perhaps be:

$$RDT_+ \sim \text{Binomial}(\gamma, RDT_{total})$$

$$\text{logit}(\gamma) = \beta_1 \text{post} + \beta_2 \text{high. travel} + \beta_3 \text{post} \times \text{high. travel} + \beta_4 \mathbf{X} + \varepsilon$$

Here, the probability gamma is estimated assuming the binomial data, given positive (RDT₊) cases out of all (RDT_{total}) tested and reported individuals. The explanations in the text (with the exception of β_2 see below) hold, but here, β_4 is a vector given the covariates in matrix \mathbf{X} that represent the different covariates included in the respective models. The unexplained error is described by (ε). In the edited manuscript neither β_4 or ε are currently explained. Please add this.

We have added explanations for these variables in the manuscript.

The variable “stratum” is a value of 1 or 2 and is also used to make the survey design object in the github code. What is this term in the context of this work? The survey package description notes “For multistage sampling the id argument should specify a formula with the cluster identifiers at each stage. If subsequent stages are stratified “strata” should also be specified as a formula with stratum identifiers at each stage.”

As described in the methods (line 277), the stratum variable is assigned to each enumeration area based on population density and estimated local residual transmission. Stratum 1 are areas with higher LRT and lower population density, while stratum 2 are areas with higher population density and lower LRT. We added this additional explanation into the methods to clarify for the reader.

The strata dictated how many households in the EA were sampled during the survey. As such, it was used to calculate the weights, and indicated as the stratum variable in our survey design.

The id variable was the EA (psuID) and then the household ID, as households were sampled within the enumeration area.

The edited manuscript describes β_2 as the difference in prevalence of infections between low- and high- travel in 2019, but is this not the difference between low and high travel at any time?

The β_2 variable is correctly described in the manuscript, and is only the difference between high and low travel areas in 2019. To get differences in 2020, one would need to take the linear combination of β_2 and β_3 (the interaction term). If we disregard covariates, and focus on the simple linear model:

$$\text{Pr}(RDT_+) = \beta_0 + \beta_1 \text{POST} + \beta_2 \text{hightravel} + \beta_3 \text{POST} * \text{hightravel}$$

In which POST = 0 for the year 2019 and 1 for 2020, and hightravel (ht) = 1 for high travel areas, and 0 for low travel areas, the prevalence estimate for each year and travel area is as follows:

$$\text{Low travel 2019 (POST} = 0, \text{ht} = 0) = \beta_0 + \beta_1 * 0 + \beta_2 * 0 + \beta_3 * 0 * 0 = \beta_0$$

$$\text{High travel 2019 (POST} = 0, \text{ht} = 1) = \beta_0 + \beta_1 * 0 + \beta_2 * 1 + \beta_3 * 0 * 1 = \beta_0 + \beta_2$$

$$\text{Low travel 2020 (POST} = 1, \text{ht} = 0) = \beta_0 + \beta_1 * 1 + \beta_2 * 0 + \beta_3 * 1 * 0 = \beta_0 + \beta_1$$

$$\text{High travel 2020 (POST} = 1, \text{ht} = 1) = \beta_0 + \beta_1 * 1 + \beta_2 * 1 + \beta_3 * 1 * 1 = \beta_0 + \beta_1 + \beta_2 + \beta_3$$

As such, β_2 is the result of subtracting low travel 2019 from high travel 2019. In 2020, the result of high travel – low travel is $\beta_2 + \beta_3$.

Second, it would be very useful to overlay the prevalence points for the MIS figure (Figure 2) for each year. Could you also note whether these surveys are repeated in sentinel villages or are villages randomly chosen to represent an Island average? Then you can comment on any limitations. The uncertainty bars on Figure 2 look fairly consistent so plotting the points on top for the respective surveys could be a good way to visualise how well these represent the MIS data. You could jitter points and stagger them for the green and yellow lines.

We have added the points by EA to Figure 2 as suggested.

Since 2015, the survey has been island wide, in which all sectors were included in the sampling frame (Prior to 2019- sampling was done on a community level and required communities to have >20 households, but almost all were included). Households within enumeration areas are chosen. Prior to 2019, the sampling within each EA was done proportional to size. Since 2019, it was done based on the stratum, and all households were eligible for inclusion.

Finally, we thank the authors for diligently addressing the points raised during the first review. The manuscript was very clearly written previously and now contains a couple of rushed sections – these are added to address reviewer comments. We suggest tightening writing in these sections to ensure typos are eliminated and points are not repeated. But all necessary details are there, and it is an interesting approach worthy of publication.

Thank you for this comment. We have gone back through and attempted to fix typos and tighten up some of the longer sections added in response to reviewer inquiries while still leaving the information requested.